# Therapeutic Drug Monitoring in Arterial Hypertension

**DOI:** 10.3390/jpm13050815

**Published:** 2023-05-11

**Authors:** Sergey Seleznev, Alexey Shchulkin, Pavel Mylnikov, Elena Yakusheva, Natalia Nikulina

**Affiliations:** 1Department of Hospital Therapy with a Course of Medical and Social Expertise, Ryazan State Medical University, Ryazan 390026, Russia; 2Department of Pharmacology, Ryazan State Medical University, Ryazan 390026, Russia

**Keywords:** arterial hypertension, antihypertensive drugs, HPLC MS/MS

## Abstract

(1) Background: This study was planned to assess the concentration of antihypertensive drugs (AHD) in the blood serum in patients with controlled and uncontrolled arterial hypertension (AH). (2) Methods: We assessed 46 patients with AH. Based on the results of 24 h blood pressure monitoring (ABPM), the patients were randomized into two groups. The first group consisted of the patients with controlled AH; the second group consisted of the patients with uncontrolled AH. Venous blood was taken in both groups of patients in the morning before and 2 h after taking drugs to assess the concentration of lisinopril, amlodipine, valsartan, and indapamide. (3) Results. The first group included 27 patients, and the second group 19 patients. In patients with uncontrolled AH, the median concentrations of lisinopril, indapamide, amlodipine, and valsartan before and after taking the drugs did not differ from patients who reached the target BP values. (*p* > 0.05). In some patients with uncontrolled and controlled (shown for the first time) AH the concentration of AHD was below the limit of quantitative determination. (4) Conclusions. The obtained results indicate that the pharmacokinetics of AHD, apparently, does not play a significant role in the development of ineffectiveness of the ongoing therapy for AH. Therapeutic drug monitoring can be used to test adherence to the treatment.

## 1. Introduction

Arterial hypertension (AH) is one of the most common diseases. According to the K.T. Mills’ systematic analysis, in 2010 the prevalence of AH in the world was 31.1% [1], according to the ESSE-RF study, carried out in 2017 in Russia, 44.2% of the population suffered from AH [2].

In recent years despite the evidence-based approach to the treatment of AH, including lifestyle modification and 5 main groups of antihypertensive drugs (AHD), as well as starting combination therapy, it is not always possible to achieve target blood pressure (BP) in all cases.

Resistant hypertension (RHTN) is defined as BP that is uncontrolled despite using ≥3 medications of different classes, commonly a long-acting calcium channel blocker, a blocker of the renin–angiotensin system (angiotensin-converting enzyme inhibitor, or angiotensin receptor blocker) and a diuretic. All agents should be administered at maximum or maximally tolerated doses and at the appropriate dosing frequency [3].

RHTN is generally attributable to persistent fluid retention [4], secondary, in large part, to hyperaldosteronism and chronic excessive sodium ingestion [5], and also hyperactivation of the renin-angiotensin-aldosterone system [6], sympathoadrenal system [7], modification of AHD targets [8].

The refractory hypertension (RfHTN) phenotype refers to patients whose BP remains uncontrolled despite the use of maximal or near-maximal antihypertensive therapy. 

RfHTN is defined as BP that is uncontrolled despite using ≥5 different antihypertensive agents, including a long-acting thiazide diuretic (i.e., chlorthalidone) and mineralocorticoid receptor antagonists (spironolactone or eplerenone) [9].

Most uncontrolled AH is not truly resistant to medical treatment but results from factors that lead to or maintain elevated BP readings independent of prescribed pharmacological treatment, termed pseudoresistance. The most common causes of pseudo-RHTN are inaccurate BP measurement, resulting in falsely elevated readings, the white coat effect, where in-office BP is persistently elevated but out-of-office BP is at goal, undertreatment, including clinical inertia, and medication nonadherence. Identification of factors that contribute to pseudoresistance is important in preventing costly and potentially risky diagnostic evaluations of patients who are not truly resistant to treatment and avoiding inappropriate intensification of treatment, which can be costly and potentially increases the risk of adverse events [10]. 

The prevalence of RHTN ranges from 5 to 30% based on the definition used by relevant studies [11]. However, the true prevalence of RHTN after applying a strict definition and having excluded causes of pseudo-RHTN is less than 10% of the patients with treated AH. Importantly, RHTN is related to a higher risk for cardiovascular morbidity and mortality, chronic kidney disease, and other AH-mediated target organ damage [12]. Thus, the development of new approaches to the treatment of RHTN is an important medical problem. Progress in pharmacology over the past 50 years has clearly shown that the concentration of many drugs in the blood correlates with their pharmacological activity. Thus, concentration is a good candidate than dosage to quantify the efficacy of the ongoing therapy or toxicity. Therapeutic drug monitoring (TDM) is the clinical practice of measuring the concentration of drugs in blood or plasma or other body fluids that may be correlated with the concentration of substances in the blood. This measured drug concentration can be used afterward to adjust the drug dosing regimen by achieving a given concentration or exposure interval, called the therapeutic range [13].

In cardiology, TDM has been developed for single drugs. For example, it has been shown that high blood serum concentration of digoxin in patients with congestive heart failure was associated with an increased overall mortality rate from all causes (0.5–0.8 ng/mL, 29.9%; 0.9–1.1 ng/mL, 38.8%, and > or =1.2 ng/mL, 48.0%, *p* = 0.006). In patients with digoxin concentration between 0.5 and 0.8 ng/mL the mortality rate was 6.3% (95% confidence interval (CI), 2.1–10.5%) lower compared to patients receiving placebo [14].

However, there are only a few studies in which TDM was performed in patients with AH in order to assess patients’ adherence to treatment [15,16,17]. 

Evaluation of the dependence of the effectiveness of antihypertensive therapy on the concentration of AHD in the blood has not been currently carried out. Herewith, it is logical to assume that the decrease of the concentration of AHD in blood below the minimum effective one can also make a significant contribution to the ineffectiveness of the therapy. This study was aimed at testing this hypothesis. 

The object of this study was to test the concentration of AHD in patients with uncontrolled and for the first time in patients with controlled AH.

## 2. Materials and Methods

We performed a clinical one-step controlled study on the basis of the Ryazan Regional Clinical Cardiological Dispensary (Ryazan, Russia), the study period covered February 2022–February 2023.

Inclusion Criteria:Age over 18;Signed informed consent form;An established diagnosis of AH based on the Clinical Guidelines “Arterial hypertension in adults”, approved by the Scientific and Practical Council of the Ministry of Health of the Russian Federation, 2020 [18];Mandatory patient compliance with recommendations for lifestyle modification in accordance with the Clinical Guidelines “Arterial hypertension in adults”, approved by the Scientific and Practical Council of the Ministry of Health of the Russian Federation, 2020 [18].Regular administration of any two AHD (lisinopril, amlodipine, valsartan) in combination with indapamide for a month, possibly in fixed combinations, in stable dosages;Fertile female patients must use proper methods of contraception throughout the study period.

Exclusion Criteria:Patient’s connection with the organization or conducting of the study;Pregnancy.

We enrolled in our study 46 patients with AH. 

All patients underwent a routine examination, which included: anthropometry, assessment of BP, heart rate (HR), general and biochemical blood tests, urinalysis, and echocardiography.

In addition, all patients underwent 24-h blood pressure monitoring (ABPM). According to the results of ABPM, we formed two groups of patients: 

1. Controlled AH. Patients, who, according to ABPM, met the following criteria: mean daily systolic blood pressure (SBP) < 135 mm Hg, mean daily diastolic blood pressure (DBP) < 85 mm Hg, the average night SBP < 120 mm Hg, the average nighttime DBP < 80 mm Hg. 

2. Uncontrolled AH. Patients, who, according to ABPM, had at least one of the following markers of poor BP control: mean daily SBP ≥ 135 mm Hg, mean daily DBP ≥ 85 mm Hg, the average night SBP ≥ 120 mm Hg, the average nighttime DBP ≥ 80 mm Hg. 

After randomization, in the morning before the next AHD administration and 2 h after AHD administration, venous blood samples were collected to assess the concentration of lisinopril, amlodipine, valsartan, and indapamide by highly efficient liquid chromatography with tandem mass spectrometric (HPLC MS/MS) detection, using chromato-graph Ultimate 3000 and mass-spectrometer TSQ Fortis (ThermoFisher).

For sample preparation, we have used methanol containing fexofenadine at a concentration of 1 ng/mL as an internal standard, which was added to serum samples in a ratio of 3:1 (600 µL of methanol with fexofenadine and 200 µL of serum). The resulting mixture had been shaken on a shaker for 1 min, then centrifuged at 19,000× *g* (Avanti JXN-3, Beckman Coulter) for 10 min at 40 °C. 600 µL of the supernatant was pipetted into labeled vials and placed in the autosampler for further analysis.

The volume of the injected sample was 5 μL. 

Separation was performed on a UCT Selectra C18 4.6 mm × 100 mm, 3 μm, 100A column with a Selectra C18 Guard Cartridges SLC-18GDC46-3UM at a temperature of 35 °C, in a gradient elution mode in a ratio of 0.1% formic acid solution/acetonitrile: 0 min-80%/20%, 0.1 min-60%/40%, 6 min-15%/85%, 10 min-15%/95%, 10 min-80%/20% with a flow rate of 400 µL/min and positive detection electrospray ionization mode, electrospray voltage 3500 V, sheath gas 50 arb, auxiliary gas 10 arb, sweet gas 1 arb, vaporizer temperature 350 °C, ion transport tube 300 °C, using multiple reaction monitoring (MRM) modes with speed argon supply 2 mTorr. 

All the parameters for detection for each target compound were presented in Table 1.

The laboratory method was validated for the following parameters: selectivity, calibration curve, accuracy, precision, limit of quantitation, sample transfer, sample stability, and matrix effect [19].

The linearity was obtained by preparing different calibration curves within the concentration of 1–1000 ng/mL. The calibration curves showed good linearity with correlation coefficients (R^2^ ≥ 0.99), ranging from 1 to 1000 ng/mL (Table 2).

Precision and accuracy studies were performed by spiking matrix-spiked samples at different quality control (QC) levels (1, 3, 500, and 1000 ng/mL) of each analyte, and five replicates were analyzed at each concentration on each of three different days. 

Data regarding the intra- and inter-day precision of the analysis of samples are shown in Table 3. Both the intra- and inter-day assay results were within the acceptable variability ranges. These values were within acceptable limits (i.e., 20% for the lower limit of quantification and 15% for other concentrations), indicating that this analysis method was reproducible and reliable in human serum samples.

Extraction recovery and matrix effect were determined for (QC) samples from the serum at two concentrations (3 and 1000 ng/mL) (Table 4). These results suggest that the simple protein precipitation method is suitable for the efficient extraction of tested drugs from human serum.

In the present study, we assessed the analyte’s stability in terms of long-term storage (90 days), autosampler (24 h), and freeze/thaw conditions using two different QC samples (3 and 1000 ng/mL); the precision of the quantification method for the analyte was determined to be within 15%. 

The results obtained were processed with the use of StatSoft Statistica 13.0 (USA, license number JPZ811I521319AR25ACD-W) and Microsoft Excel for MAC ver. 16.24 (ID 02984-001-000001). The distribution of the obtained data was evaluated by the Shapiro-Wilk test. If the value had normal distribution, we used the Student’s T test to assess the statistical significance of differences. In other cases, the Mann°Whitney test was used. Frequency values were compared with the Chi-square test.

The results obtained are presented in tables and graphs as the mean and standard deviation (M (SD)) in data with a normal data distribution or median, minimum and maximum values, upper and lower quartiles (Me (min; max) or Me (Q1; Q2)) for non-normal distribution.

## 3. Results

Among 46 patients enrolled in our study, there were 19 men (41%). The first group of patients with controlled AH included 27 patients, and the second group-19 patients.

Demographic data are presented in Table 5.

Among the 46 patients included in the study, there were 19 men (41%). The first group of patients with controlled AH included 27 patients, and the second group 19 patients.

It is noteworthy that among the patients with uncontrolled AH, there were statistically significantly more men. At the same time, when we compared the groups by age, and BMI, no statistically significant differences were found.

Information about comorbidities is presented in Table 6. Statistically significant differences in the incidence of diseases such as diabetes mellitus, angina pectoris, old myocardial infarction, stroke, and gastrointestinal diseases, that could affect metabolism AHD, were not detected.

The analysis of the laboratory data also did not reveal any statistically significant differences in fasting glucose levels, kidney function, lipid metabolism, electrolytes, and liver damage (Table 7).

The data of 24 h BP monitoring are presented in Table 8.

Based on the study inclusion criteria, BP values were statistically significantly higher in the group of uncontrolled AH, while there were no statistically significant differences in AHD doses (Table 9).

While studying the concentrations of AHD in the blood serum, the following results were obtained (Table 10).

In patients with uncontrolled AH, the median steady-state concentrations (before taking the drug) of indapamide, amlodipine, and valsartan did not differ statistically significantly from those of patients who reached the target BP.

Meanwhile, the equilibrium concentration of lisinopril in patients with uncontrolled AH was higher than in patients with controlled AH at the borderline of statistical significance (*p* = 0.06). After recalculating these indicators for the dose and body weight of the patients, no statistically significant differences in the concentration of AHD were obtained (Table 10).

Two hours after taking all the studied AHD, there were no significant differences in the medians of their concentrations between the groups of controlled and uncontrolled AH (*p* > 0.05).

When calculating the increase in the concentrations of the studied AHD in the blood serum 2 h after their administration (that is, the delta of changes in concentrations), the following results were obtained (Table 11). 

The increase in the concentrations of lisinopril, valsartan and amlodipine did not differ statistically significant between the compared groups.

At the same time, the delta of change of indapamide concentration in the group of patients with Uncontrolled AH exceeded the delta of change in the group of Controlled AH at the borderline of statistical significance (*p* = 0.058).

After recalculating these indicators for the dose and body weight of the patients, no statistically significant differences in the delta of change in serum concentrations of AHD were obtained.

It is interesting to note, that in several patients with controlled AH, the concentration of AHD was below the lower limit of quantification (lisinopril 26%, amlodipine 22%, indapamide 22%, valsartan 50%). In the group of patients with uncontrolled AH, in some cases, AHD was also not found in the blood (lisinopril 11%, amlodipine 32%, indapamide 37%) (Figure 1).

The difference between the proportion of the patients in the group with controlled and uncontrolled AH, in whose blood at least one AHD was not found, was not statistically significant (*p* = 0.318). These results indicate that, regardless of BP control, at least 30% of patients with AH, taking AHD (lisinopril, amlodipine, valsartan, indapamide), do not achieve therapeutic concentrations, which is a potential opportunity to increase the effectiveness of the therapy.

Additionally, it was found out, that in the blood serum of three patients with BP control, there were AHD, which were not prescribed to the patients, and in the group of patients with uncontrolled AH there was one such recipient. These results indicate that despite the treatment in a hospital, some patients took medications without a doctor’s prescription.

## 4. Discussion

Optimal pharmacological treatment of AH is of great importance in reducing the incidence of cardiovascular events and kidney diseases [20,21]. 

Failure to achieve BP targets despite the use of ≥3 AHD is defined as RHTN. The pathogenesis of RHTN is multifactorial [22].

In the framework of this study, the contribution of AHD pharmacokinetics abnormality in the ineffectiveness of the ongoing therapy for AH was assessed.

For the tested drugs, it was found that their effect depends on their concentration in the blood serum [23,24,25].

Concentrations of lisinopril, indapamide, amlodipine, and valsartan in serum were detected by the original and validated HPLC-MS/MS method.

Our study has shown that in patients with uncontrolled AH, the concentration of AHD was not less than in patients with controlled AH. Lisinopril concentration had a pronounced tendency to increase in the group of uncontrolled AH. The revealed differences are most likely correlated with a higher dose of AHD, taken by the patients with uncontrolled AH, as evidenced by the leveling of differences when recalculating the concentration of AHD per dose.

There were no significant differences in AHD concentrations between the groups of controlled and uncontrolled AH two hours after taking them.

The maximum concentration is reached after two hours of taking only for indapamide 2.5 mg [23] and valsartan [26]. 

The maximum concentration of lisinopril is reached after 6.2 ± 0.2 h [24], amlodipine after 6–8 h [25], and indapamide 1.5 mg after 11 ± 7 h [27]. That is, 2 h after taking amlodipine, lisinopril did not reach the maximum concentration. However, after 2 h, these drugs caused a decrease in blood pressure [24,25].

The lack of concentration analysis of tested drugs 6–8 h after ingestion is a limitation of the present study.

Our results have made evident, that the concentration of AHD does not seem to play a significant role in the ineffectiveness of the treatment of AH.

In most patients concentrations of all tested AHD were within the therapeutic range: for lisinopril 1–140 ng/mL, for amlodipine 5–18 ng/mL, for valsartan 800–6000 ng/mL [28], for indapamide 25–75 ng/mL [27].

However, approximately 30% of patients had at least one analyte below the lower limit of quantitation, including in patients with controlled hypertension (shown for the first time). 

These results are consistent with similar results for assessing adherence to treatment in patients with arterial hypertension.

A recent study assessed medication adherence in patients with RfHTN by measuring AHD and their metabolites in 24 h urine specimens using high-performance liquid chromatography-tandem mass spectrometry. Of 40 patients with RfHTN, only 16 (40%) had complete adherence with all prescribed medications, 18 (45%) had partial adherence, and 6 (15%) were completely non-adherent with all prescribed medications. Overall, 21 (52.5%) were adherent with 5 or more medications, including chlorthalidone and a mineralocorticoid receptor antagonist [29]. This study shows that only about half of patients with RfHTN are adherent to all of their medications as assessed by urine drug and drug metabolite levels. A study in Germany assessed adherence in patients with RHTN, defined as BP ≥ 140/90 mm Hg and/or 24 h ABPM ≥ 130/80 mm Hg despite the use of 3 or more medications, including a diuretic, using detection of antihypertensive medications or their metabolites in urine. The results showed similar adherence to that of patients with RfHTN: 47.4% were adherent to all medications, 37% were adherent to some of the prescribed medications, and 15.8% took none of the prescribed medications [30].

On the other hand, a decrease in AHD concentrations may be a consequence of a change in their pharmacokinetics: a decrease in absorption in the gastrointestinal tract, an increase in biotransformation, and an acceleration of excretion, which, in turn, may be due to the peculiarities of the genetic status of the patients (polymorphism of genes encoding biotransformation enzymes (for example, families of cytochromes Z450) or transporter proteins (for example, P-glycoprotein)), as well as drug-drug interactions [31].

Another factor that could affect the rate of absorption of antihypertensive drugs is the quality of the dosage forms [32].

In our study, all patients took the same drugs (only generics). Therefore, this factor did not affect the obtained results.

## 5. Conclusions

In summary, this study confirmed, that blood serum concentrations of AHD (lisinopril, indapamide, amlodipine, valsartan) do not differ in patients with controlled and uncontrolled AH. The obtained results indicated that the concentration of AHD does not play a significant role in the ineffectiveness of the ongoing therapy of AH.

In some patients with uncontrolled and controlled (shown for the first time) AH the concentration of AHD was below the limit of quantitative determination.

Thus, the approach to the treatment and diagnosis of patients with uncontrolled arterial hypertension should be comprehensive, including an assessment of the physiological, genetic, and biochemical factors of patients, and therapeutic drug monitoring can be used to assess adherence to the treatment.

## Figures and Tables

**Figure 1 jpm-13-00815-f001:**
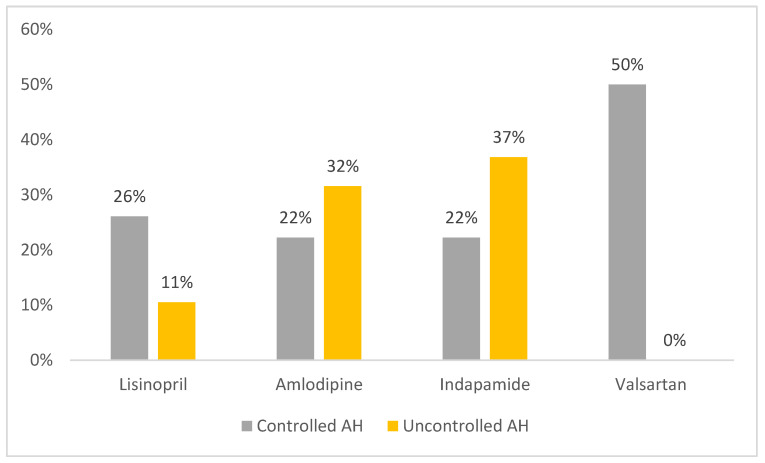
The proportion of patients with AHD serum concentrations below the limit of quantitation.

**Table 1 jpm-13-00815-t001:** Optimized MRM conditions.

Drug	Precursor, *m*/*z*	Product, *m*/*z*	Collision Energy, V	Tube Lens, V
indapamide	365.8	117	36	111
	365.8	131.4 *	15	111
lisinopril	405.85	84 *	30	112
	405.85	245.4	24	112
amlodipine	409.2	237.8 *	12	100
	409.2	293.9	10	100
valsartan	436.2	206.3 *	27	104
	436.2	234.9	18	104
fexofenadine	502.3	171	27	110
	502.3	466.2 *	27	110

*—product ion mass used for quantification of the analyte.

**Table 2 jpm-13-00815-t002:** Regression equation of lisinopril, amlodipine, indapamide and valsartan.

Drug	Calibration Curve 1	Calibration Curve 2	Calibration Curve 3
Indapamide	y = 0.0000327611 + 0.0000288278*x, R^2^ = 0.9987	y = 0.0000198026 + 0.0000439326*x, R^2^ = 0.9985	y = 0.000052618 + 0.0000411263*x, R^2^ = 0.9975
Lisinopril	y = 0.0000359385 + 0.0000421611*x, R^2^ = 0.9996	y = 0.000047645 + 0.00004347*x, R^2^ = 0.9999	y = 0.0000460434 + 0.0000433505*x, R^2^ = 0.9992
Amlodipine	y = 0.000112708 + 0.000139241*x, R^2^ = 0.9997	y = 0.000118841 + 0.000138061*x, R^2^ = 0.9940	y = 0.00004.19279 + 0.000138121*x, R^2^ = 0.9997
Valsartan	y = −0.0000040368 + 0.0000309164*x, R^2^ = 0.9991	y = −0.000000280674 + 0.0000260166*x, R^2^ = 0.9997	y = −0.0000126283 + 0.0000205256*x, R^2^ = 0.9988

**Table 3 jpm-13-00815-t003:** Accuracy and precision data for indapamide, lisinopril, amlodipine and valsartan.

Drug	QQ, ng/mL	Intra-Day Accuracy, %, *n* = 5	Intra-Day Precision, %, *n* = 5	Inter-Day Accuracy, %, *n* = 15	Inter-Day Precision, %, *n* = 15
Indapamide	1	4.64	11.45	2.69	13.02
	3	3.4	13.39	3.39	10.09
	500	14.08	0.422	2.86	7.63
	1000	14.22	0.53	6.62	5.65
Lisinopril	1	1.46	10.76	5.91	12.19
	3	0.45	8.06	6.03	7.64
	500	2.97	4.64	0.02	5.16
	1000	0.45	8.19	1.49	5.86
Amlodipine	1	16.1	5.35	5.29	15.13
	3	10.53	10.23	4.52	10.36
	500	3.01	2.78	2.21	2.72
	1000	5.41	2.09	3.41	3.99
Valsartan	1	5.66	10.58	0.82	13.91
	3	3.47	10.01	2.64	9.28
	500	3.99	10.98	5.41	9.97
	1000	2.72	8.93	8.34	7.35

**Table 4 jpm-13-00815-t004:** Recovery and matrix effect data for indapamide, lisinopril, amlodipine and valsartan.

Drug	Concentration, ng/mL	Matrix Effect, %	Recovery, %
Indapamide	3	78.95 ± 8.65	82.25 ± 10.84
	1000	91.40 ± 2.72	71.07 ± 10.27
Lisinopril	3	135.03 ± 10.07	112.75 ± 4.87
	1000	111.86 ± 12.66	106.79 ± 10.29
Amlodipine	3	78.59 ± 7.84	64.49 ± 8.56
	1000	93.25 ± 8.16	80.84 ± 5.61
Valsartan	3	78.32 ± 9.70	72.62 ± 7.89
	1000	89.21 ± 8.45	70.45 ± 3.29
fexofenadine	1	134.85 ± 8.09	111.46 ± 6.32

**Table 5 jpm-13-00815-t005:** Demography of patients enrolled in the study.

Value	All Patients (*n* = 46)	Controlled AH (*n* = 27)	Uncontrolled AH (*n* = 19)	*p*
N	46	27	19	
Sex	f-27, m-19	f-20, m-7	f-7, m-12	0.016
Age, y (M (SD))	65.1 (10.4)	66.3 (9.5)	63.4 (11.5)	0.35
BMI, kg/m^2^ (M (SD))	31.4 (5.28)	31.0 (4.6)	31.8 (6.2)	0.64

**Table 6 jpm-13-00815-t006:** Concomitant diseases.

Concomitant Disease	All Patients		Controlled AH		Uncontrolled AH		*p*
	N	%	N	%	N	%	
Angina	28	61%	20	43%	8	17%	1
PCI	1	2%	1	2%	0	0%	1
Stroke	3	7%	2	4%	1	2%	1
Old myocardial infarction	4	9%	3	7%	1	2%	1
Atrial Fibrillation/Flutter	15	33%	9	20%	6	13%	0.29
Diabetes	10	22%	6	13%	4	9%	0.43
Aortic aneurism	1	2%	0	0%	1	2%	1
Chronic gastritis	7	15%	6	13%	1	2%	0.65
Chronic cholecystitis	4	9%	3	7%	1	2%	1
Chronic pancreatitis	3	7%	3	7%	0	0%	1

**Table 7 jpm-13-00815-t007:** Laboratory results.

Value	All Patients, Me (Q1; Q3)	Controlled AH, Me (Q1; Q3)	Uncontrolled AH, Me (Q1; Q3)	*p*
Fasting glucose, mmol/L	6.4 (5.6; 6.8)	6.4 (5.6; 6.8)	6.1 (5.6; 6.7)	0.37
Urea, mmol/L	6.1 (5.1; 7.3)	6.1 (5.2; 7.4)	5.6 (4.7; 7.3)	0.62
Serum creatinine, μmol/L	81.0 (64.6; 98.6)	74.5 (63.8; 93.9)	85.1 (69.9; 103.4)	0.21
Creatinine clearance, mL/min	90.3 (69.9; 103.2)	87.2 (61.4; 99.8)	95.7 (78.6; 118.0)	0.20
Uric acid, mmol/L	322.7 (271.6; 387.4)	300 (254.8; 378.7)	349.7 (309.9; 428.5)	0.12
Total bilirubine, μmol/L	12.7 (10.1; 15.9)	12.9 (10.9; 15.8)	12.6 (9.9; 15.9)	0.89
AsT, IU/L	20.4 (16.4; 24.4)	21.5 (17.7; 23.2)	17.9 (15.5; 25.9)	0.16
AlT, IU/L	22.6 (18.5; 30.0)	23.2 (19.7; 27.8)	19.5 (18.3; 38.5)	0.73
Total cholesterol, mmol/L	5.1 (4.2; 6.0)	4.8 (4.21; 6.0)	5.6 (4.1; 6.0)	0.74
LDL cholesterol, mmol/L	3.2 (2.0; 3.9)	3.0 (1.8; 3.9)	3.8 (2.0; 4.0)	0.38
HDL cholesterol, mmol/L	1.2 (1.1; 1.4)	1.2 (1.1 1.4)	1.1 (1.0; 1.4)	0.53
Triglycerides, mmol/L	1.6 (1.2; 2.0)	1.4 (1.1; 2.0)	1.7 (1.4; 2.1)	0.39
Potassium, mmol/L	4.4 (4.1; 4.7)	4.31 (4.1; 4.6)	4.5 (4.2; 4.8)	0.55
Sodium, mmol/L	142.2 (141.3; 143.2)	142.4 (141.0; 143.8)	142.2 (141.7; 143.0)	0.65

**Table 8 jpm-13-00815-t008:** The 24 h BP monitoring data.

Value	All Patients, M (SD)	Controlled AH, M (SD)	Uncontrolled AH, M (SD)	*p*
SBP, day, mm Hg	131.4 (17.6)	121.3 (9.5)	145.9 (16.3)	<0.001
DBP, day, mm Hg	74.6 (13.1)	68.7 (8.9)	84.4 (13.9)	<0.001
SBP, night, mm Hg	121.3 (17.9)	109.4 (8.6)	137.5 (15.1)	<0.001
DBP, night, mm Hg	65.1 (11.1)	58.1 (5.6)	74.9 (9.4)	<0.001

**Table 9 jpm-13-00815-t009:** Doses of AHDs.

Value	All Patients, Me (Q1; Q3)	Controlled AH, Me (Q1; Q3)	Uncontrolled AH, Me (Q1; Q3)	*p*
Lisinopril, dose, mg	20.0 (40.0; 40.0)	20.0 (5.0; 40.0)	20.0 (10.0; 40.0)	0.12
Amlodipine, dose, mg	5.0 (2.5; 20.0)	5.0 (2.5; 10.0)	10.0 (5.0; 10.0)	0.28
Indapamide, dose, mg	2.5 (1.5; 2.5)	2.5 (1.5; 2.5)	2.5 (1.5; 2.5)	0.578
Valsartan, dose, mg	160.0 (80.0; 160.0)	160.0 (160.0; 160.0)	160.0 (160.0; 160.0)	1.0

**Table 10 jpm-13-00815-t010:** Serum concentrations of AHD.

Value	Controlled AH, Me (Min; Max)	Uncontrolled AH, Me (Min; Max)	*p*
Lisinopril, 0 h, ng/mL	23.6 (0.0; 375.1)	98.3 (0.0; 362.9)	0.06
Lisinopril, 2 h, ng/mL	94.2 (0.0; 274.9)	107.5 (0.0; 581.2)	0.5
Lisinopril, 0 h, dose conversion, ng/mL/mg	4.7 (0.0; 37.5)	10.2 (0.0; 36.3)	0.095
Lisinopril, 2 h, dose conversion, ng/mL/mg	5.6 (0.0; 14.1)	5.2 (0.0; 47.5)	0.29
Lisinopril, 0 h, conversion to body weight, ng/mL/kg	0.5 (0.0; 4.3)	1.0 (0.0; 4.8)	0.19
Lisinopril 2 h, conversion to body weight, ng/mL/kg	1.8 (0.0; 3.6)	1.5 (0.0; 6.3)	0.94
Amlodipine, 0 h, ng/mL	6.97 (0.0; 26.55)	8.2 (0.0; 16.5)	0.28
Amlodipine, 2 h, ng/mL	8.3 (0.0; 23.4)	9.47 (0.0; 16.5)	0.52
Amlodipine, 0 h, dose conversion, ng/mL/mg	0.9 (0.0; 5.3)	0.8 (0.0; 3.3)	0.50
Amlodipine, 2 h, dose conversion, ng/mL/mg	1.1 (0.0; 4.7)	0.8 (0.0; 3.2)	0.72
Amlodipine, 0 h, conversion to body weight, ng/mL/kg	0.1 (0.0; 0.3)	0.06 (0.0; 0.2)	0.14
Amlodipine, 2 h, conversion to body weight, ng/mL/kg	0.1 (0.0; 0.3)	0.07 (0.0; 0.2)	0.24
Indapamide, 0 h, ng/mL	13.4 (0.0; 103.0)	12.3 (0.0; 102.0)	0.35
Indapamide, 2 h, ng/mL	12.4 (0.0; 102.0)	21.1 (0.0; 102.0)	0.12
Indapamide, 0 h, dose conversion, ng/mL/mg	5.5 (0.0; 21.4)	3.89 (0.0; 10.97)	0.42
Indapamide, 2 h, dose conversion, ng/mL/mg	6.6 (0.0; 39.2)	10.4 (0.0; 31.1)	0.19
Indapamide, 0 h, conversion to body weight, ng/mL/kg	0.1 (0.0; 0.4)	0.1 (0.0; 0.3)	0.14
Indapamide, 2 h, conversion to body weight, ng/mL/kg	0.1 (0.0; 1.0)	0.2 (0.0; 1.0)	0.32
Valsartan, 0 h, ng/mL	294.2 (0.0; 521.3)	611.9 (488.12; 735.77)	0.19
Valsartan, 2 h, ng/mL	608.6 (174.1; 1353.9)	1463.5 (1317.7; 1609.4)	0.26
Valsartan, 0 h, dose conversion, ng/mL/mg	1.8 (0.0;3.2)	3.0 (3.0; 3.0)	1.0
Valsartan, 2 h, dose conversion, ng/mL/mg	6.9 (2.2; 101.5)	8.2 (8.2; 8.2)	1.0
Valsartan, 0 h, conversion to body weight, ng/mL/kg	1.7 (0.0; 5.3)	3.8 (3.8; 3.8)	1.0
Valsartan, 2 h, conversion to body weight, ng/mL/kg	0.07 (0.02; 45.4)	0.06 (0.06; 0.06)	1.0

**Table 11 jpm-13-00815-t011:** Delta of change in serum concentrations of AHD.

Value	Controlled AH, ng/mL, Me (Min; Max)	Uncontrolled AH, ng/mL, Me (Min; Max)	*p*
Lisinopril	71.8 (−117.5; 207.4)	50.7 (−24.6; 265.1)	0.16
Amlodipine	0.9 (−6.9; 8.5)	2.2 (−6.5; 10.8)	0.39
Indapamide	4.58 (−25.0; 55.2)	16.3 (−3.0; 54.2)	0.058
Valsartan	430.8 (−86.2; 859.1)	829.5 (829.5; 829.5)	1.0

## Data Availability

The data presented in this study are available on reasonable request from the corresponding author.

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
