# Peer review of "Therapeutic Drug Monitoring in Arterial Hypertension"

_jpm, 2023, doi:10.3390/jpm13050815_

Round 1

Reviewer 1 Report

The study should include in the introduction not only the pharmacological background that supports the lack of efficacy of the antihypertensive drug (physiological, pharmacological and pharmaceutical).

It should include clearer background information to support its objective.

In methodology, it should include the ethics committee that supported the study, as well as a letter of informed consent.

In the presentation of results, it should omit conclusive elements, and it should also include in a very general way the results of the validation of the method used (this is the one that denotes the certainty of quantitative analysis).

In discussion it should include the physiological and genetic elements that can explain the alteration of the concentration, it is also important to consider the manufacture of the drug (pharmaceutical criteria: degree of compression of the formulation, particle size, uniformity of content, among others).

Restate the conclusion with the physiological, genetic and pharmaceutical background.

Author Response

Dear Reviewer!

Point 1. The study should include in the introduction not only the pharmacological background that supports the lack of efficacy of the antihypertensive drug (physiological, pharmacological and pharmaceutical).

Response 1: we have extended the Introduction

Point 2. It should include clearer background information to support its objective.

Response 2: background information was updated.

Point 3. In methodology, it should include the ethics committee that supported the study, as well as a letter of informed consent.

Response 3: we have included the ethics committee that supported the study, as well as a letter of informed consent

Point 4. In the presentation of results, it should omit conclusive elements, and it should also include in a very general way the results of the validation of the method used (this is the one that denotes the certainty of quantitative analysis).

Response 4: we corrected the presentation of results

Point 5. In discussion it should include the physiological and genetic elements that can explain the alteration of the concentration, it is also important to consider the manufacture of the drug (pharmaceutical criteria: degree of compression of the formulation, particle size, uniformity of content, among others).

Response 5: we have extended the Discussion and conclusions

Point 6. Restate the conclusion with the physiological, genetic and pharmaceutical background.

Response 6: the conclusion was restated with the physiological and pharmaceutical background. A possible genetic aspects of uncontrolled blood pressure were not included in the article, because this is a separate area of research, the results will be published later

Many thanks and king regards from the authors

Reviewer 2 Report

The authors describe a trial looking at the advantages of TDM to monitor antihypertensive therapy with four different drugs. As the monitoring of blood pressure is very cheap, non invasive and can be done easily the motivation for this study can not be identified by the reviewer. This correlates then well with the conclusion of the authors that the pharmacokinetics of these antihypertensive drugs does not play a significant role in the ineffectiveness of therapy.

- Abstract, lines 17 - 19: the sentence is not readable

- Introduction, lines 39f: This statement of the authors is wrong. There are many drugs for which it has been shown extensively that there is no correlation between drug concentration in blood and pharmacological effect

- Material and methods, line 64: the study period is in the future (February 22 - February 24). Do the results describe science fiction?

- Material and methods, line 98: What is the evidence to take the delta concentration between trough and the 2 hour after intake of the drug concentration? Does the 2h value represent the maximal concentration?

- Material and methods, line 103f: for an optimal LC-MS methode deuterated standards should be used. Why have the authors selected fexofenadine?

- Results: has the LC-MS method been published before? If not the results of the validation must be described.

- Results, Table 3: This data do not add anything to the content of the manusccript.

- Results Table 6: the concentrations cannot be correct for the converted data. I assume that the concentration is ng/ml /mg dose for the dose conversion and ng/ml /k BW for the body weight conversion

- Results, line 197: which are the therapeutic concentrations of the four antihypertensive drugs? The authors should either show or cite, that there is a correlation between drug concentration and pharmacological effect.

Author Response

Dear Reviewer!

Point 1:  The authors describe a trial looking at the advantages of TDM to monitor antihypertensive therapy with four different drugs. As the monitoring of blood pressure is very cheap, non invasive and can be done easily the motivation for this study can not be identified by the reviewer. This correlates then well with the conclusion of the authors that the pharmacokinetics of these antihypertensive drugs does not play a significant role in the ineffectiveness of therapy.

Response 1: we extended the Introduction

Point 2: Abstract, lines 17 - 19: the sentence is not readable

Response 2: we corrected abstract

Point 3: Introduction, lines 39f: This statement of the authors is wrong. There are many drugs for which it has been shown extensively that there is no correlation between drug concentration in blood and pharmacological effect

Response 3: we extended the Introduction

Point 4: Material and methods, line 64: the study period is in the future (February 22 - February 24). Do the results describe science fiction?

Response 4: we have corrected study period

Point 5: Material and methods, line 98: What is the evidence to take the delta concentration between trough and the 2 hour after intake of the drug concentration? Does the 2h value represent the maximal concentration?

Response 5:  we analyzed drug concentrations at 2h as this is the Tmax for valsartan and indapamide. For amlodipine Tmax 6 h, lisinopril 7 h

Point 6: Material and methods, line 103f: for an optimal LC-MS methode deuterated standards should be used. Why have the authors selected fexofenadine?

Response 6: unfortunately, we were not able to buy deuterated standards. fexofenadine has intermediate physicochemical properties among all analyzed substances and showed good metrological characteristics

Point 7: Results: has the LC-MS method been published before? If not the results of the validation must be described.

Response 7: We described validation in Material and methods

Point 8: Results, Table 3: This data do not add anything to the content of the manusccript.

Response 8: We have not deleted Table 3. This table represents the information about liver and kidney function of both patients’ group. Liver and kidney functions can affect on antihypertensive drugs methabolism

Point 9: Results Table 6: the concentrations cannot be correct for the converted data. I assume that the concentration is ng/ml /mg dose for the dose conversion and ng/ml /k BW for the body weight conversion

Response 9: We corrected Table 6

Point 10: Results, line 197: which are the therapeutic concentrations of the four antihypertensive drugs? The authors should either show or cite, that there is a correlation between drug concentration and pharmacological effect.

Response 10: we described the therapeutic concentrations of the four antihypertensive drugs in Discussion

Thank you very much for your work on reviewing our article

Round 2

Reviewer 2 Report

The authors have take the recommendation of the reviewer into account and the manuscript has improved tremendously. The explain the time of blood sampling in the answert to the reviewer, but do not add this information into the manuscript. This should be done. As the 2h concentration represents the maximal concentration only for valsartan and indapamide and not for amlodipine and lisinopril the sampling time and its influence on the results should also be discussed in the Discussion section of the manuscript. For the last two drugs the time of sampling is still in the time range where the concentration of the drug is increasing.

Author Response

Dear Reviewer!

We have updated the manuscript according to your remarks

Best regards,

Sergey Seleznev